# Deciphering the Complex Relationships Between the Hemostasis System and Infective Endocarditis

**DOI:** 10.3390/jcm14113965

**Published:** 2025-06-04

**Authors:** Muhammad Aamir Wahab, Atta Ullah Khan, Silvia Mercadante, Iolanda Cafarella, Lorenzo Bertolino, Emanuele Durante-Mangoni

**Affiliations:** 1Department of Precision Medicine, University of Campania “Luigi Vanvitelli”, 80138 Naples, Italy; muhammadaamir.wahab@unicampania.it (M.A.W.); ataullah.khan@unicampania.it (A.U.K.); emanuele.durante@unicampania.it (E.D.-M.); 2Department of Advanced Medical and Surgical Sciences, University of Campania “Luigi Vanvitelli”, 80138 Naples, Italy; silvia.mercadante@libero.it (S.M.); iolanda.cafarella95@gmail.com (I.C.)

**Keywords:** infective endocarditis, hemostasis, thrombophilia, immunothrombosis, embolic risk, antiplatelet therapy, anticoagulation

## Abstract

Infective endocarditis (IE) arises from complex interactions between microbial pathogens and host hemostasis systems, where dysregulated coagulation mediates microbial persistence and systemic thromboembolic complications. Alterations in primary, secondary, and tertiary hemostasis in the acute IE phase have direct clinical implications for vegetation formation and detachment. *Staphylococcus aureus* is one of the most common pathogens that causes IE, and it is capable of profoundly altering the coagulation cascade through several mechanisms, such as platelet activation, prothrombin activation through staphylocoagulase release, and plasminogen stimulation via staphylokinase production. Understanding these complex and yet unmasked mechanisms is of pivotal importance to promoting targeted therapeutic intervention aimed at reducing IE morbidity and mortality. Moreover, the management of antiplatelet and anticoagulant treatment during IE onset is a controversial issue and needs to be tailored to patient comorbidities and IE-related complications, such as cerebral embolism. This review provides a roadmap to promote clinicians’ understanding of the complex interactions between hemostasis and IE clinical manifestations and complications, discussing pathogen-specific coagulation profiles while addressing critical knowledge gaps for IE management.

## 1. Introduction

Human blood harbors an intricate balance between fluidity and clotting, a delicate equilibrium finely regulated by the hemostasis system. The defense mechanisms of the human body operate at a high level of biological complexity, with the hemostasis system standing as a critical guardian against potential pathological and physiological disruptions [1,2]. This fundamental biological mechanism encompasses the strict interplay of cellular and molecular components working in a synchronized manner to protect the body from potential hemorrhagic and thrombotic complications [3,4,5].

The hemostasis function comprises three interconnected phases: primary, secondary, and tertiary hemostasis [1,6] (Table 1). Primary hemostasis rapidly forms a platelet plug at the site of vascular injury and is characterized by platelet adhesion, activation, and aggregation, and this remarkably complex initial response involves approximately 6000 mRNA species responsible for encoding receptors, ion channels, signaling molecules, and regulatory proteins [6]. Secondary hemostasis, also called coagulation, involves a cascade of molecular interactions responsible for initial platelet plug stabilization, and this process involves the activation of intrinsic and extrinsic pathways [7,8], resulting in the formation of thrombin, a key enzyme of clot formation [6]. The coagulation mechanism is regulated via positive and negative feedback loops, which ensure proper control over the clotting processes [6,8]. Tertiary hemostasis, alternatively called fibrinolysis, functions as a key regulatory mechanism preventing excessive clot formation via the proteolytic degradation of the fibrin network, resulting in the release of clotting components into the bloodstream [9]. The process is regulated by plasminogen activators responsible for the conversion of plasminogen into plasmin, which degrades fibrin, maintaining vascular patency and preventing potential thrombotic complications [6]. Evidence has accumulated to suggest that all three phases of the hemostasis function may be significantly altered in infective endocarditis (IE) (Table 1).

Indeed, IE is a complex thrombo-inflammatory disease of the endocardium in which the hemostasis system becomes both a defender and a potential accomplice in disease progression [10]. This complex pathogenesis emerges from the dynamic interplay between pathogenic microorganisms and host hemostasis system responses, with recent studies indicating mortality rates of 17–20% during hospitalization and 30–35% at one year [11,12,13,14]. IE development needs a complex convergence of multiple factors, such as cardiac valve surface alteration, with experimental studies demonstrating that normal valvular endothelium tends to be resistant to bacterial colonization and that damage to the valve surface provides bacteria a suitable attachment site [15]. The occurrence of bacteremia with invasive potential, such as that due to specific bacterial species like *Staphylococcus aureus*, *S. epidermidis*, and *Enterococci* having the ability to adhere to cardiac tissues, readily results in the formation of infected “vegetation” through microbial growth within the protective matrix of fibrin and platelets [15,16].

IE pathogenesis is centered upon the formation of vegetation, where bacteria become embedded within a fibrin mesh together with platelets and inflammatory cells. Certain pathogens, particularly *S. aureus*, show remarkable abilities to manipulate the hemostasis system via the formation of thrombin-like activity through staphylocoagulase and von Willebrand factor-binding proteins [17,18,19]. Additionally, key hemostasis parameters serve as crucial prognostic indicators in IE, and previous research has demonstrated that elevated D-dimers and prolonged PT-INR are associated with higher in-hospital mortality, while prolonged aPTT correlates with increased one-year mortality [13]. Moreover, higher D-dimer levels and shorter aPTT are significantly associated with embolic complications, particularly in *S. aureus* infections [13]. Understanding the role of the hemostasis system in IE is crucial for several reasons, including improving risk stratification and prognostic accuracy. The activity of the hemostasis system is highly relevant in terms of the susceptibility, progression, and treatment of IE [10]. Recent investigations have demonstrated that monitoring coagulation parameters can provide valuable insights into disease progression and potential complications.

Given the critical role of hemostasis in IE pathogenesis and its implications for patient management, a thorough review of our current understanding of this relationship is essential. This article synthesizes current knowledge and recent findings from basic science and clinical research to explore the interplay between hemostasis and IE. By clarifying these relationships, we aim to identify potential therapeutic targets and improve strategies for managing the delicate balance between thrombosis and bleeding in patients with IE, aiming to provide both the practicing physician and the involved scientist with a comprehensive overview of the hemostasis system in the context of IE. In addition, we explore the interactions between bacterial pathogens and the host’s hemostatic mechanisms, identifying the clinical implications of hemostatic disturbances in managing IE and evaluating potential therapeutic interventions targeting hemostasis in affected patients.

## 2. Current Understanding of Infective Endocarditis Pathogenesis

The pathogenesis of IE is a complex process involving interactions between endothelial damage, hemostasis mechanisms, microbial virulence factors, and the host’s immune responses, in which endocardial damage and platelet–fibrin deposition play crucial roles.

### 2.1. Endocardial Damage and Platelet–Fibrin Deposition

Endocardial injury occurs through degenerative, inflammatory, and mechanical pathways that predispose cardiac valves to thrombotic complications. Hemodynamic stresses from congenital defects or acquired valve abnormalities generate pathological shear forces through turbulent flow patterns, disrupting endothelial integrity via mechano-transduction pathways involving notch signaling and the activation of flow-sensitive transcriptional regulators [20,21,22,23]. These mechanical insults synergize with age-related valvular degeneration, characterized by the remodeling of the extracellular matrix and calcific processes that further compromise endothelial barrier function [24,25,26]. The inflammatory component involves cytokine cascades—particularly Interleukin 1 beta (IL-1β) and Tumor Necrosis Factor alpha (TNF-α)—which exacerbate endothelial dysfunction through the NF-κB-mediated upregulation of adhesion molecules and matrix metalloproteinases [27,28,29,30]. These cytokines originate from infiltrating leukocytes and activated endothelium during systemic inflammatory states, creating feedforward loops that accelerate valvular deterioration [31,32]. Modern iatrogenic risks include endothelial disruption from intravascular and intracardiac devices, where catheter/device surfaces promote platelet adhesion through VWF-mediated mechanisms [33], while substances like illicit drugs induce direct toxic damage via adrenergic overstimulation and oxidative stress pathways [34].

Furthermore, after the endothelial breach, the exposed subendothelial collagen and VWF trigger platelet adhesion via GPIb-IX-V receptors, while tissue factor exposure initiates thrombin-mediated fibrin deposition [35,36]. This sterile platelet–fibrin matrix serves as a bacterial substrate through three key adhesion mechanisms: direct matrix binding, facilitated by fibronectin via microbial surface components recognizing adhesive matrix molecules (MSCRAMMs), fibrinogen bridging via clumping factors (ClfA, ClfB), and plasma protein intermediaries mediated by VWF-binding protein (vWbp) by *S. aureus* and *Streptococci*.

Transient bacteremia seeds these thrombi with *S. aureus*, employing vWbp to bind VWF under shear stress, while coagulase activity generates fibrin–platelet microthrombi that enhance bacterial retention [34]. This adhesion cascade explains the clinical progression from endothelial injury to IE through sequential sterile vegetation formation and microbial colonization [35,37].

### 2.2. Infected Vegetation Growth

The pathogenesis of infected vegetation in IE initiates with the bacterial colonization of damaged endocardial surfaces [38]. *S. aureus* reaches an extraordinary density of 10^10^–10^11^ CFU/g in the valve tissue/vegetation interface via balancing replication with metabolic adaptation, resulting in stratified microenvironments where the bacteria on the biofilm surface are metabolically active while the core populations are in a dormant state to evade immune detection [39,40,41]. Interestingly, vegetations beyond 10 mm in diameter correlate with a 3.2-fold increase in embolization risk and an elevated 30-day mortality [42,43,44].

Molecular studies reveal that *S. aureus* vegetation-forming strains downregulate the RNAIII, sarA, and sigB regulatory systems [45,46,47], resulting in elevated surface adhesin expression (MSCRAMMs/SERAMs) for endothelial attachment [48,49], promote sequential transitions from colonization factors to the production of exotoxin [50,51], and facilitate protease-rich biofilm development, conferring resistance to phagocytosis [52]. The proteolytic landscape within the vegetation mediates pathogenesis via fibronectin/collagen degradation [47,53], complement protein cleavage (C3, C5a) [46], and clotting factor processing that drives fibrin deposition [54,55] mechanisms potentiated by high bacterial protease activity [52].

Staphylococcal vegetations exhibit distinctive pathobiological characteristics compared to non-staphylococcal strains. They show a considerably greater proteomic uniformity (with a coefficient of variation below 15%, in contrast to the 25–40% range observed in non-staphylococcal infections) and a strikingly consistent pattern of fibrinogen integration, markedly unlike the pathogen-specific variability seen in streptococcal or enterococcal vegetations [56]. Unlike non-staphylococcal strains, which rely on multiple exoenzymes specific to their environments [57,58] the staphylococcal strains typically possess many coagulases and superantigens such as SEC and TSST-1 [47,59,60,61]. There is also a difference in their immune evasion strategies: staphylococcal strains use biofilm matrices [39,62], while non-staphylococcal strains use different capsular polysaccharides [39,63,64]. This structural complexity shields the bacteria from the host’s defenses, allowing them to keep multiplying.

### 2.3. Vegetation-Borne Complications: Embolization and Septic Emboli

IE represents a significant clinical challenge, with systemic embolic complications that occur in 21–50% of cases and contribute substantially to one-year mortality rates of 15–30% in treated patients [43,65]. The dislodgement of infected vegetation initiates a pathogenic cascade where circulating emboli obstruct vascular flow and trigger systemic inflammation, exacerbating endothelial damage through cytokine-mediated mechanisms [65,66]. Vegetation characteristics critically influence embolic risk, with lesions exceeding 10 mm demonstrating three-fold more significant embolic potential than smaller formations [43]. Mitral valve vegetations, particularly those on the anterior leaflet, show heightened embolization propensity due to increased mechanical stress during ventricular systole [65]. Morphological features further modulate risk, with mobile, filiform vegetations exhibiting 40–60% embolic rates versus 15–20% for sessile variants [67]. Younger patients and those with higher C-reactive protein (CRP) levels show higher propensities for embolic complications [68,69].

Microbial etiology significantly impacts embolic dynamics. *S. aureus* etiology poses a high embolic risk through rapid vegetation growth and friable architecture, while *S. viridans* typically form slow-growing and dense vegetations with delayed embolic manifestations [65,70] (Table 2).

Embolic patterns diverge by valve involvement: left-sided IE predominantly causes cerebral (60%), splenic (25%), and renal (15%) emboli through systemic circulation, whereas right-sided IE manifests primarily as septic pulmonary emboli (85%), frequently progressing to infarction (40%) or empyema (15%) [65,73]. Contemporary risk stratification integrates advanced imaging modalities with biomarker profiling. Four-dimensional cardiac CT demonstrates a sensitivity of 96% and a specificity of 97% for detecting vegetations compared with surgical findings [74].

Critical reappraisal of the available evidence allows for the risk stratification of patients in terms of their likelihood of developing embolic complications, as detailed in Table 3.

## 3. Influence of Infective Endocarditis on Hemostasis System Function

### 3.1. Platelet Pre-Activation in IE

In *S. aureus* IE, platelet activation is driven by surface proteins and host receptors. The pathogen binds directly to platelets via iron-responsive surface determinant B (IsdB), which interacts with glycoprotein IIb/IIIa (GPIIb/IIIa) [86,87,88,89], while clumping factors A/B (ClfA/B) utilize fibrinogen or fibronectin as bridges to the same receptor [87,90,91] (Figure 1). Plasma IgG further strengthens adhesion by linking staphylococcal protein A to platelet FcγRIIa [90,92,93,94]. Bacterial toxins exacerbate activation: α-toxin forms membrane pores, triggering calcium influx and granule secretion [90,95]; staphylococcal superantigen-like 5 (SSL-5) binds GPIbα and GPVI to induce the release of pro-inflammatory mediators [92,96]; and staphopain A, a cysteine protease, activates αIIbβ3 integrins and promotes P-selectin exposure to facilitate aggregation [95].

Platelets release antimicrobial agents like thrombocidins and β-defensin-1 (hBD-1) from α-granules to combat *S. aureus*, but bacterial resistance mechanisms enable their survival within platelet aggregates. Complement proteins modulate interactions, with C1q-coated bacteria binding platelet gC1q-R [97,98,99,100] and C3b linking to P-selectin to enhance pathogen clearance while amplifying inflammation [97,101,102]. Activated platelets recruit neutrophils, generating neutrophil extracellular traps (NETs) in vegetation [99]. Although NETs limit bacterial spread, they worsen tissue damage and vegetation growth [103,104]. Furthermore, platelet reactivity varies, with some patients developing hypocoagulable profiles due to consumption coagulopathy [85,93,105]. Persistent platelet activation also promotes biofilm formation, reducing antibiotic efficacy [106].

In sepsis, platelet heterogeneity and increased activation significantly influence coagulation and immune response. Despite lower overall counts, an increased fraction of activated platelets, often thrombin-mediated, drives a procoagulant state. Specific platelet subpopulations, like the fatal cluster C4, exhibit high activity in coagulation and hemostasis pathways; notably, genes enriched in C4 are linked to bacterial endocarditis and thrombosis. These activated platelets also contribute to endotheliopathy and disseminated intravascular coagulation by releasing procoagulant molecules, which are critical processes in both severe sepsis and the development of vegetations in IE [107,108].

Emerging therapies that could target these mechanisms include GPIIb/IIIa antagonists to disrupt adhesion [109,110], SSL-5 inhibitors to block toxin effects, and staphopain A inhibitors to reduce protease-driven aggregation [95,96]. None of them have been tested in clinical trials or are currently approved for use in IE.

### 3.2. Coagulation Cascade Activation and Hypercoagulability Due to IE

Coagulation cascade activation and the resultant hypercoagulability are central pathophysiological hallmarks of IE, arising from the complex interplay between microbial virulence factors and the host’s hemostasis system, facilitating vegetation formation and thromboembolic complications [16,85]. Bacterial colonization causes significant endothelial damage, exposing subendothelial collagen and von Willebrand factor (VWF), which initiates platelet adhesion. Additionally, *S. aureus* clumping factors A/B and other surface proteins from pathogens bind to fibrinogen and platelet receptors, leading to aggregation and the formation of the fibrin–platelet matrix necessary for vegetation development [16,111,112]. Activated monocytes and damaged endothelium trigger thrombin generation through the extrinsic pathway by expressing tissue factor (TF). Bacterial lipopolysaccharides, along with peptidoglycan, activate factor XII, further promoting coagulation through the contact system [16,17,113]. This dual-pathway activation creates a self-reinforcing cycle, with thrombin increasing fibrin deposition, activating platelets, and elevating TF expression, resulting in vegetation growth rates correlating with thrombin–antithrombin complex levels [114]. This procoagulant state is evidenced by elevated levels of coagulation markers such as prothrombin fragments 1 + 2, thrombin–antithrombin complexes, and D-dimer in patients with IE [85,114,115]. The persistent activation of coagulation in IE can lead to consumption coagulopathy, potentially progressing to disseminated intravascular coagulation in severe cases [116]. Moreover, the hypercoagulable state in IE contributes to the elevated risk of thromboembolic events, a significant cause of morbidity and mortality in these patients [114].

### 3.3. Hemostasis and Innate Immunity Interaction (Immunothrombosis) in IE

In IE, the interplay between hemostasis and innate immunity—termed immunothrombosis—has a paradoxical role in disease progression. Bacterial pathogens, particularly *S. aureus*, exploit coagulation pathways in order to adhere to damaged endocardial surfaces, where fibrin and platelet aggregates form protective vegetation to shield bacteria from immune clearance [17,93]. Activation of the coagulation cascade occurs through both extrinsic (tissue factor-driven) and intrinsic (factor XII-mediated) pathways, triggered by bacterial components like cell wall elements and nucleic acids [16,17]. Thrombin, a central enzyme in this process, generates fibrin and amplifies inflammation by interacting with platelet thrombin receptors and modulating leukocyte recruitment [17,117,118] (Figure 2). Platelets contribute beyond clot formation by releasing antimicrobial peptides and facilitating neutrophil extracellular trap (NET) formation, yet their activation paradoxically boosts vegetation maturation and supports bacterial persistence [92,93]. Meanwhile, bacterial pathogens hijack fibrinolytic mechanisms, by stimulating plasminogen activation and increasing fibrin degradation [17,119,120] (Figure 3). This dysregulated immunothrombotic response creates a cycle of valve destruction, embolic complications, and systemic inflammation, while therapeutic targeting remains challenging due to bleeding risks associated with anticoagulant or antiplatelet therapies [16,17,121]. Recent studies highlight a non-linear relationship between platelet counts and mortality in IE, underscoring the delicate balance between thrombotic containment and pathological clot formation in this life-threatening infection [17,121].

Recent advances in immunothrombosis modulation in IE have centered on dual strategies of NETosis inhibition and intrinsic coagulation blockade. Peptidylarginine deiminase 4 (PAD4) inhibitors, including GSK484 and Cl-amidine, attenuate histone citrullination and NET formation, while in *S. aureus* endocarditis models, PAD4 inhibition and DNase-I-mediated extracellular DNA degradation significantly reduce vegetation size and bacterial load [104,122]. Concurrently, targeting coagulation factor XI via the monoclonal antibody abelacimab and oral small-molecule inhibitor milvexian disrupts contact-activation-driven thrombin generation with a minimal impact on hemostasis, as shown by reduced postoperative venous thromboembolism in knee arthroplasty trials [123,124]. Emerging FXI inhibitors such as MK-2060 and asundexian are under clinical evaluation across cardiovascular and thromboembolic indications, underscoring the translational potential of combined NETosis and FXI inhibition to modulate immunothrombosis in infective endocarditis [125].

## 4. Role of Thrombophilia in Infective Endocarditis

### 4.1. Definition and Types of Thrombophilia: Inherited and Acquired

Thrombophilia encompasses a spectrum of disorders characterized by an increased propensity for venous or arterial thrombosis due to imbalances in procoagulant and anticoagulant factors [126,127,128]. It is broadly categorized into inherited (hereditary) and acquired forms, each with distinct etiologies and pathophysiological mechanisms. Inherited thrombophilia arises from genetic mutations affecting key regulatory proteins in the coagulation cascade. The most prevalent forms include Factor V (FV) Leiden mutation (G1691A), present in 3–8% of European and U.S. populations, which confers resistance to activated protein C [126,129]. FII mutation (G20210A), occurring in 1.7–3% of the same populations, leads to elevated prothrombin levels [126]. Furthermore, deficiencies in natural anticoagulants, including antithrombin, protein C, and protein S, impair thrombin regulation and fibrinolysis [126,127,130]. These defects typically follow autosomal dominant inheritance, though severe protein C/S deficiencies may exhibit recessive patterns [126,127].

In contrast, acquired thrombophilia results from non-genetic factors, such as antiphospholipid syndrome (APS), marked by lupus anticoagulant, anticardiolipin, or anti-β2-glycoprotein I antibodies, which induces a hypercoagulable state through platelet activation and endothelial dysfunction [131]. In myeloproliferative neoplasms (MPNs) and paroxysmal nocturnal hemoglobinuria (PNH), clonal mutations (e.g., JAK2 V617F) or GPI-anchor deficiencies promote thrombo-inflammation [128]. Moreover, several other conditions may act as secondary triggers, such as malignancy, surgery, oral contraceptives, pregnancy, or chronic inflammation, which elevate clotting factors (e.g., factor VIII) or reduce anticoagulant synthesis [126,131]. While hereditary forms often manifest as unprovoked venous thromboembolism (VTE) in younger individuals, clinical expression depends on gene-environment interactions, with most thrombotic events requiring additional acquired risk factors [127,132,133]. Acquired thrombophilias frequently present in adulthood and may resolve with treatment of underlying conditions, for instance, immunosuppression for APS [128]. Contemporary guidelines emphasize thrombophilia testing only when results would directly alter clinical management, such as anticoagulation duration or family counseling [128,132].

### 4.2. Specific Thrombophilic Conditions Associated with Infective Endocarditis

Thrombophilia, in general, influences IE outcomes, and, more specifically, inherited thrombophilias affect the clinical trajectory of IE, with distinct patterns observed across device-related and native valve infections; the FVL and FII G20210A mutations are 2-fold more prevalent in IE patients compared to healthy populations (6.4% vs. 3.25%; OR 2.03; *p* = 0.047), with FVL disproportionately linked to device-related IE and FII mutations to prosthetic valve IE (allele frequency 8.3% vs. 2.2% in native valve IE; *p* = 0.021) [81]. These genetic variants enhance thrombus-mediated bacterial adhesion onto damaged endocardium or device surfaces, creating niches for infection. While neither mutation directly correlates with vegetation size or embolic risk [81], patients with thrombophilias exhibit a trend toward higher in-hospital mortality (OR 1.8; *p* = 0.08), likely due to synergistic microvascular thrombosis and impaired pathogen clearance [81].

In addition, acquired thrombophilias further modulate outcomes via APS, accelerating the formation of vegetation through β2-glycoprotein-I-mediated platelet activation and complement dysregulation, contributing towards increasing embolic risk [134,135]. Moreover, malignancy-associated hypercoagulability predisposes individuals to NBTE, resulting in a fibrin-rich substrate for secondary infection [81,136], and COVID-19-related immune-thrombosis intensifies endothelial damage and NETosis, increasing the risk of IE in critically ill patients [136].

Furthermore, device-related IE demonstrates stronger associations with inherited hypercoagulable states, as FVL’s interaction with intravascular hardware amplifies fibrin deposition and biofilm formation [81,137]. In contrast, native valve IE shows no significant thrombophilia enrichment beyond baseline population rates, suggesting divergent thrombogenic mechanisms [81,138]. Prosthetic valve IE patients with FII mutations face heightened thrombotic complications, potentially exacerbating valve dysfunction and systemic embolization [81]. The prognostic impact of thrombophilias extends beyond acute infection, with carriers exhibiting prolonged inflammatory markers and elevated D-dimer levels post-treatment [13]. This persistent hypercoagulable state may contribute to delayed healing and recurrent thromboembolic events. Current evidence underscores the need for thrombophilia screening in IE patients with recurrent device infections or atypical microbiological profiles, as 24% carry unknown hypercoagulable states requiring tailored anticoagulation strategies [81,139]. However, therapeutic anticoagulation in thrombophilic IE fails to reduce ischemic events (OR 1.10; *p* = 0.37) while increasing hemorrhagic risk (OR 1.51; *p* = 0.03) [140], emphasizing the necessity for genotype-guided management [81].

The current understanding of the potential role of thrombophilia in IE is summarized in Table 4.

### 4.3. The Potential Role of Thrombophilia in IE: A Cause or a Consequence?

The interplay between thrombophilia and IE reveals a complex bidirectional relationship where thrombophilia acts both as a predisposing factor and a consequence of IE pathogenesis [17,81]. Inherited thrombophilias such as the FVL and FII G20210A mutations were found to be more prevalent in IE patients compared to the controls, suggesting a potential role in facilitating and increasing early vegetation formation through increased fibrin deposition at the sites of endothelial injury [81]. However, these genetic variants show no direct correlation with vegetation size or embolic risk, indicating that thrombophilia’s contribution may be limited to initial susceptibility rather than disease progression; conversely, IE drives a pathological immune-thrombotic cascade where bacterial pathogens (notably *S. aureus*) activate platelets, induce NETs, and achieve control of the coagulation pathways to form vegetations that are fibrin-rich [16,17,147]. This infection-triggered hypercoagulable state amplifies systemic embolization risk, accounting for as many as 20–40% of IE patients suffering from complications like stroke and peripheral emboli [16,84]. Pathogens such as *S. aureus* further exploit this environment by secreting coagulases that directly activate FII, embedding bacteria within protected thrombo-inflammatory niches [17] (Figure 2). Observational data suggest that inherited thrombophilias modestly increase IE risk [81], while clinical thrombotic events predominantly arise from infection-mediated coagulation activation rather than pre-existing hypercoagulability [16,84]. This is supported by the limited efficacy of chronic antiplatelet therapy in reducing embolic events, despite its lowering of mortality [16]. Thus, thrombophilia in IE represents both a minor predisposing factor in genetically susceptible individuals and a major downstream effect of pathogen-driven immunothrombosis [16,17].

## 5. Diagnostic and Therapeutic Implications

### 5.1. Effect of Antibiotic Therapy on IE Vegetation and Embolic Risk

The timing of antibiotic therapy is critical in modulating embolic risk in IE. About 65% of embolic events occur during the first 2 weeks after antibiotic treatment starts, equal to a 10- to 20-fold higher embolic risk immediately post-treatment, as compared to later phases, when such risk dramatically declines [75,148] (Table 5).

This embolic “vulnerability” correlates with vegetation dynamics, which were shown to paradoxically change based on the actual antibiotics given: vancomycin achieved a 45% size reduction, whereas cephalosporins were associated with a 40% increase in vegetation size [149,152]. Effective antimicrobial regimens that reduce vegetation size by ≥40% substantially lower embolic rates [149,152,153]. The combination of early surgery and antibiotics demonstrated improved outcomes for vegetations exceeding 10 mm in size (OR 2.28 for embolism) [43], decreasing mortality risk substantially by 78% (HR 0.22) compared to the administration of medical therapy alone [151] (Table 5).

Valve surgery within 48 h of IE diagnosis in patients with mobile mitral valve vegetations translates into a significant reduction in the embolic risk, preventing about 22% of embolic strokes [154]. The EASE trial compared early surgery (37 patients) with conventional treatment (39 patients) for left-sided IE characterized by severe valve disease and large vegetations. The primary endpoint—a composite of in-hospital death and embolic events within 6 weeks—occurred in 3% of the early-surgery group compared to 23% of the conventional group (hazard ratio 0.10, *p* = 0.03). At 6 months, all-cause mortality rates were similar (3% for early surgery vs. 5% for conventional, *p* = 0.59), but the composite endpoint of death, embolic events, or recurrence was significantly lower in the early-surgery group (3% vs. 28%, hazard ratio 0.08, *p* = 0.02). Early surgery significantly reduces the risk of systemic embolism in this patient population [155].

The overall risk profile is significantly modulated by microbial factors, where *S. aureus* infections are associated with 35–61% embolic rates [75,76] while Streptococcus bovis and fungal pathogens independently increase the likelihood of embolism, possibly enhancing the propensity of vegetation parts to detach and generate emboli/fragments that detach and become emboli [65,78]. These organism-specific risks underscore the necessity of accurate pathogen-directed antimicrobial selection, particularly given staphylococcal infections’ association with enlarging vegetation during treatment [75,76].

### 5.2. Effect of Prior or De Novo Antiplatelet Treatment on IE Vegetation and Embolic Risk

The timing and duration of antiplatelet therapy, combined with IE pathophysiology, result in divergent outcomes (Table 5). Preclinical models indicate that thromboxane A_2_ suppression through aspirin inhibits *S. aureus*-induced platelet aggregation, resulting in a reduction in bacterial density in the vegetation by 0.8 log_10_ CFU/g and of vegetation mass by 37% compared to controls [16,156]. This antiplatelet effect is responsible for the disruption of the fibrin–platelet matrices necessary for the microbial colonization of the valvular endocardium [156].

However, clinical translation indicates a striking contrast, where prior chronic antiplatelet therapy for ≥6 months before IE diagnosis correlates with an embolic risk reduction of about 64% (aOR 0.36, 95% CI 0.19–0.68) [83], while de novo initiation during active IE lacks efficacy [150]. A meta-analysis including 12,151 IE patients confirmed that chronic antiplatelet therapy resulted in a decrease in systemic thromboembolism (OR 0.53, 95% CI 0.38–0.72) without increasing the risk of intracranial hemorrhage (OR 0.35, 95% CI 0.11–1.10) [84]. Mechanistically, the vegetation could be stabilized by long-term aspirin use, which, through cyclooxygenase-1 inhibition, would alter the release of platelet-derived growth factor, resulting in decreased vegetation friability [156]. This protective effect is evident primarily in non-hypertensive patients without prior cardiovascular disease (embolic rate 8.3% vs. 28.6% in comorbid subjects) [83].

Contrastingly, randomized trials of de novo aspirin (325 mg/day) initiated after IE diagnosis show no embolic reduction (OR 1.62, 95% CI 0.68–3.86) at the expense of a 92% higher bleeding risk (OR 1.92, 95% CI 0.76–4.86) [150]. The MATIE trial revealed that delayed aspirin treatment initiation (~34 days post-symptom-onset) failed to decrease IE embolic events in patients (28.3% vs. 20.0% in placebo; OR 1.62; *p* = 0.29) while showing a trend towards an increased bleeding risk (OR 1.92; *p* = 0.075) [16,150]. This temporal discordance suggests that antiplatelets require prolonged pre-exposure to modulate endothelial–platelet interactions before vegetation formation [156]. The 2023 European Society of Cardiology (ESC) guidelines strongly advise against initiating antiplatelets in acute IE (Class III recommendation) but endorse continuing pre-existing therapy given its mortality benefit (90-day aOR 0.27, 95% CI 0.11–0.64) [16,84,157]. Critical analysis reveals a therapeutic window: chronic low-dose aspirin (75–325 mg/day) begun ≥6 months before IE onset reduces embolic burden through vegetation matrix stabilization, whereas acute administration disrupts hemostasis without altering established microbial biofilms [16,83,84,158]. Future research should explore P2Y_12_ inhibitors’ effects on vegetation composition and embolic risk stratification using vegetation size/shape metrics [43].

### 5.3. Effect of Prior or De Novo Anticoagulant Treatment on IE Vegetation and Embolic Risk

Prior anticoagulant therapy in IE also seems to exhibit time-dependent effects on vegetation dynamics and embolic risk (Table 5). Pre-admission anticoagulation, for instance with warfarin, correlates with reduced vegetation size (>10 mm) and mobility at diagnosis, potentially lowering early embolic events by limiting fibrin–platelet deposition in nascent vegetations [85,86]. This protective mechanism diminishes after antibiotic initiation as bacterial virulence factors and NETs become dominant in vegetation stability, overshadowing coagulation pathways [85,86]. Meta-analyses confirm that a vegetation size >10 mm independently predicts embolic risk (OR 2.28) and mortality (OR 1.63) [43], underscoring the limited utility of post-antibiotic-therapy anticoagulation.

De novo anticoagulation in IE remains controversial. Observational data show no significant reduction in post-admission embolic events [86,159], while hemorrhagic complications—particularly intracranial hemorrhage (ICH) in *S. aureus* IE—increase by 71% [86]. A 2023 cohort study noted lower in-hospital mortality with anticoagulation but highlighted confounding factors like comorbid conditions [84]. Furthermore, the hypercoagulability in IE, driven by inflammation-induced tissue factor upregulation and protein C depletion, may attenuate anticoagulant efficacy [85]. Guidelines strongly advise against routine anticoagulation, unless mandated by the presence of prosthetic valves or atrial fibrillation, prioritizing early surgery for large (>10 mm), mobile vegetations to mitigate embolic risk [154,160]. Individualized risk stratification remains highly important for balancing thromboembolic prevention against hemorrhagic sequelae in this high-risk population [86].

Figure 4 presents a summary of treatment decisions designed to handle antiplatelet and anticoagulant medication during the acute IE onset.

### 5.4. Future Approaches to Targeting Thrombophilia and Coagulation Abnormalities in IE

Future approaches targeting thrombophilia and coagulation abnormalities evolve towards precision therapies that balance thromboprophylaxis with bleeding risk mitigation. Such approaches could also likely be applied in IE. Emerging anticoagulants targeting factor XI (FXI) and FXIa, such as abelacimab, asundexian, and milvexian, show promise in preclinical and early-phase trials by selectively inhibiting thrombosis while preserving hemostasis—a critical advantage in many conditions, including IE, where dual thromboembolic and hemorrhagic risks coexist [84,161,162]. These agents could disrupt pathologic fibrin deposition and platelet-rich vegetation growth without exacerbating IE-related bleeding, particularly in *S. aureus*-driven IE, where dysregulated immunothrombosis is central [17]. Concurrently, glycoprotein VI inhibitors, like glenzocimab, are being explored for their antiplatelet effects, which could reduce vegetation embolization while minimizing intracranial hemorrhage in IE [161,163]. Advances in point-of-care viscoelastic testing (e.g., thromboelastography) and platelet function analysis enable the real-time monitoring of hemostasis function, guiding personalized anticoagulant dosing and the timing of surgical interventions [163]. Further, biomarker-driven strategies—such as vWF multimers and fibrinolytic activity profiling—aim to identify patients who may benefit from adjunctive therapies like fibrin-targeted thrombolytics or immunomodulators [16,17]. Clinical trials are urgently needed to validate these approaches, particularly in high-risk subgroups with prosthetic valves or renal impairment, where conventional anticoagulants remain problematic [140,161].

## 6. Conclusions

The complex interplay between bacteria and the hemostasis system in IE elucidates a ubiquitous balance between defensive host mechanisms and pathogenic exploitation. Key findings indicate that the hemostasis system contributes actively to IE pathogenesis and subsequent sequelae rather than playing a passive role. Platelet activation and hypercoagulability, in addition to immunothrombosis, drive vegetation formation as well as stability; nevertheless, thromboembolic events remain a major cause of morbidity and mortality. Thrombophilia, whether acquired or inherited, emerges as a double-edged sword—predisposing to initial infection while intensifying disease progression through increased coagulation activation. Clinically, monitoring hemostasis parameters such as D-dimer and fibrinogen levels offers prognostic value, potentially guiding risk stratification and therapeutic decisions.

Despite scientific advances, the precise molecular mechanisms by which pathogens take control of hemostasis pathways to form vegetation still need to be explored, particularly in non-staphylococcal infections; the role of genetic polymorphisms in thrombophilia-related IE susceptibility requires further research, especially in the context of understudied populations. Additionally, the long-term impacts of IE-induced hemostasis dysregulation on cardiovascular health are poorly understood and require further exploration, focusing on elucidating the pathogen-specific interactions with coagulation cascades and platelet receptors, validating novel biomarkers for the early prediction of embolic risk through multi-center trials, investigating the efficacy of next-generation anticoagulants in IE-related hypercoagulability, and tracking the development of targeted therapies that have the potential to disrupt microbial adherence without compromising hemostasis balance.

The implications for IE prevention, diagnosis, and management are significant and could result in advancing clinical practices. In IE prevention, the combination of antimicrobial and antiplatelet therapies based on personalized prophylactic strategies may have potential benefits for individuals in high-risk groups, especially those with thrombophilia and recipients of prosthetic heart valves. Furthermore, concerning diagnosis, the integration of advanced imaging techniques with coagulation profiling could potentially enhance the early detection of embolic risk assessment and vegetation instability. In the case of IE management, a nuanced approach is crucial for the application of antithrombotic therapy, calling for the necessity of maintaining existing treatment regimens in patients who are already on chronic therapies while being considerate about initiating new anticoagulants during acute infection; in addition, emerging treatments that target the immune-thrombosis pathways, including platelet glycoprotein functions, deserve consideration as a potentially safer alternative to traditional anticoagulation methods. Overall, bench research translated into effective clinical applications based on collaborative multi-disciplinary research efforts is required to enhance the outcomes in this critical and life-threatening disease.

## Figures and Tables

**Figure 1 jcm-14-03965-f001:**
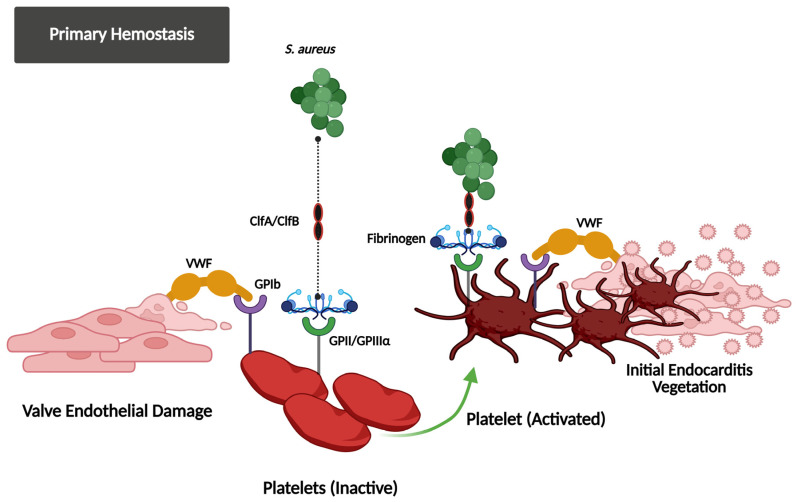
This figure illustrates how *S. aureus* exploits the primary hemostasis system to initiate IE. After valve endothelial damage, the VWF and fibrinogen bind to the exposed subendothelial matrix and facilitate platelet adhesion via the GPIb and GPIIb/IIIa receptors. *S. aureus* utilizes ClfA/ClfB adhesins to attach to these hemostasis proteins, promoting platelet activation and the formation of the initial vegetation, demonstrating the bacterial control of host hemostasis responses in the development of infection.

**Figure 2 jcm-14-03965-f002:**
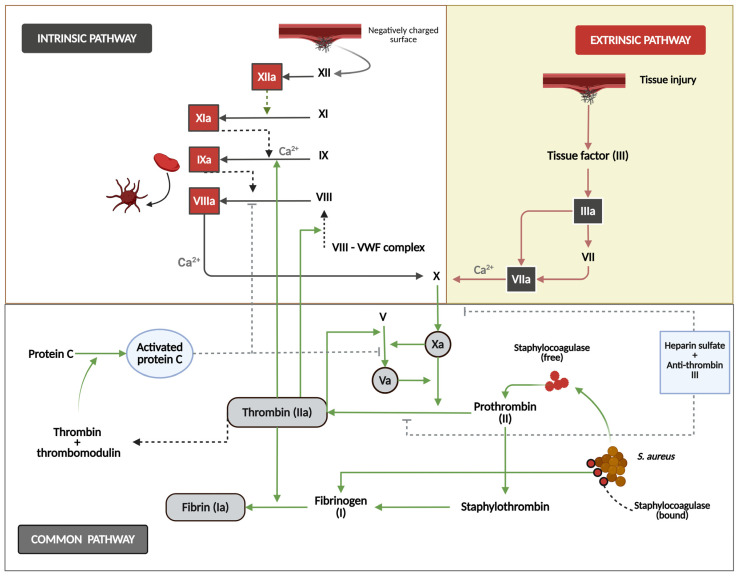
This figure shows the mechanisms of secondary hemostasis. The intrinsic pathway (left) is initiated when factor XII contacts a negatively charged surface and triggers the sequential activation of factors XI, IX, and VIII, with calcium (Ca^2+^) as an essential cofactor. The extrinsic pathway (right) starts with tissue injury that exposes tissue factor (III), activating factor VII and subsequently factor X. Both pathways converge at the common pathway in which the activated factor X converts prothrombin (FII) to thrombin (IIa), cleaving fibrinogen to form fibrin. The regulatory mechanisms include protein C, thrombin, thrombomodulin systems, heparin sulfate, and antithrombin III. The diagram also illustrates this system’s exploitation by *S. aureus* via staphylocoagulase, which activates prothrombin directly to form staphylo-thrombin, bypassing regulatory controls.

**Figure 3 jcm-14-03965-f003:**
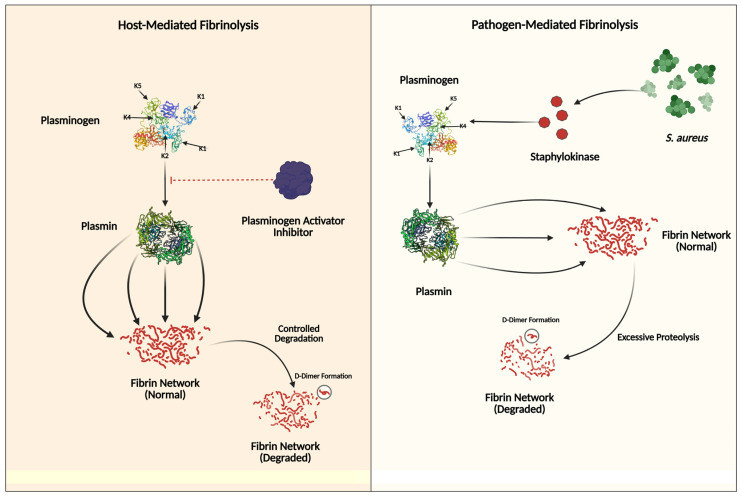
This figure illustrates the contrasting mechanisms of tertiary hemostasis (fibrinolysis regulation). The left panel shows host-mediated fibrinolysis, where plasminogen is converted to plasmin through regulated pathways involving plasminogen activator inhibitors, leading to controlled fibrin degradation and D-dimer formation. The Kringle domains (K1–K5) are crucial for plasminogen’s binding to fibrin and regulatory proteins. In the right panel, pathogen-mediated fibrinolysis demonstrates how *Staphylococcus aureus* secretes staphylokinase to activate plasminogen directly, bypassing normal inhibitory controls. This results in the excessive degradation of fibrin networks, aiding bacterial dissemination and evasion of immune responses.

**Figure 4 jcm-14-03965-f004:**
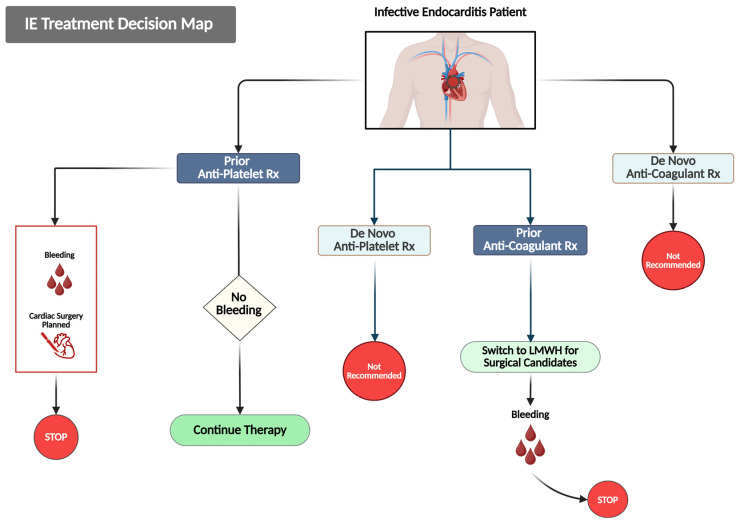
This figure outlines evidence-based decision pathways for managing antithrombotic therapy in patients with IE. Treatment approaches are categorized based on the prior and newly initiated use of antithrombotic agents. For patients already receiving antiplatelet therapy, it is advisable to continue treatment in the absence of bleeding complications. However, therapy should be halted if bleeding occurs, or if there is a planned cardiac surgery. The initiation of de novo antiplatelet therapy is not recommended. Likewise, patients on prior anticoagulation should be switched to low-molecular-weight heparin (LMWH) for the initial two weeks of treatment, and therapy should be discontinued if bleeding arises. De novo anticoagulation is also not recommended.

**Table 1 jcm-14-03965-t001:** Comparison of primary, secondary, and tertiary hemostasis in normal and IE states.

Hemostasis Phase	Components	Normal Function	Alterations in IE	Clinical Implications
**Primary Hemostasis**	Platelets, vWF (von Willebrand factor), collagen	Platelet adhesion, activation, and aggregation	Increased platelet activation, P-selectin expression	Enhanced vegetation formation, resistance to antiplatelet therapy
**Secondary Hemostasis**	Coagulation factors, thrombin	Fibrin formation and clot stabilization	Pathogen-driven activation, coagulase production	Vegetation enlargement, embolic risk
**Tertiary Hemostasis**	Plasmin, fibrinolytic enzymes	Clot dissolution and remodeling	Bacterial exploitation for tissue invasion	Compromised vegetation stability, septic emboli

**Table 2 jcm-14-03965-t002:** Pathogens and their role in IE.

Microorganism	Vegetation Characteristics	Embolic Risk/Rate	Coagulopathy Effects
***Staphylococcus aureus*** [17,71,72]	Large, friable vegetations rich in bacterial biofilms and fibrin	Particularly high risk of systemic embolization	Pronounced coagulopathy via secreted coagulases, associated with higher D-dimer levels
***Streptococcus* spp.** [65,70]	Smaller, densely adherent vegetations	Size has not been shown to influence embolic potential significantly; it follows a more indolent course	Does not activate coagulation
***Candida* spp.** [17,71]	Very large, friable vegetations	Low risk	Consumption coagulopathy

**Table 3 jcm-14-03965-t003:** Risk stratification for embolic complications in IE.

Risk Factor	Risk Effect	Evidence	Comments
**Vegetation Characteristics**			
**Size > 10 mm**	OR 2.28 (95% CI 1.71–3.05)	[42,43]	Independent predictor across multiple studies; stronger association with anterior mitral leaflet vegetations
**Size > 15 mm**	OR 2.80 (95% CI 1.97–3.98)	[44,67]	Higher risk threshold with stronger predictive value
**Mobile/filiform morphology**	40–60% risk	[67]	Compared to 15–20% with sessile vegetation
**Increasing vegetation size during therapy**	OR 3.5 (95% CI 1.9–6.4)	[75]	Dynamic assessment is more significant than single measurement
**Mitral valve location**	OR 2.1 (95% CI 1.4–3.2)	[65,68]	Especially anterior leaflet due to higher hemodynamic stress
**Aortic valve vegetation with severe regurgitation**	OR 1.8 (95% CI 1.2–2.7)	[67]	Hemodynamic factors influence embolization risk
**Microcalcifications within vegetation**	89% sensitivity for embolic prediction	[67]	Detectable on cardiac CT imaging
**Microbial Factors**			
***S. aureus* etiology**	35–61%; aOR 1.76 (95% CI 1.09–2.86)	[76,77]	Cumulative embolic incidence for total embolic events; embolic risk confined to pre-treatment phase, with no independent effect after antibiotic initiation
**Fungal pathogens**	OR 2.9 (95% CI 1.5–5.4)	[65]	Associated with larger vegetations and delayed treatment response
**Streptococcus bovis**	OR 1.7 (95% CI 1.1–2.6)	[78]	Associated with gastrointestinal malignancies
**Enterococci**	OR 1.2 (95% CI 0.8–1.8)	[79,80]	Intermediate embolic risk profile
**HACEK group organisms**	OR 1.8 (95% CI 1.1–2.9)	[16]	High biofilm formation capability
**Patient factors**			
**Younger age (<50 years)**	Negative correlation with age	[68]	Possibly related to more vigorous immune response
**CRP > 75 mg/L + D-dimer > 2500 μg/L**	82% accuracy for prediction	[76]	Combined biomarker approach improves predictive accuracy
**Procalcitonin > 0.5 ng/mL**	OR 2.0 (95% CI 1.3–3.1)	[16]	Reflects ongoing bacterial invasion and inflammation
**Thrombophilia**	OR 1.8 (*p* = 0.08)	[81]	Trend toward higher in-hospital mortality
**First two weeks of antibiotic therapy**	10–20× higher risk	[75,82]	Temporal risk clustering during early treatment phase
**Prior embolic event**	OR 2.7 (95% CI 1.9–3.8)	[43]	Strong predictor of recurrent embolism
**Pre-existing cardiovascular disease**	OR 1.5 (95% CI 1.1–2.1)	[83]	Modifies protective effect of antiplatelet therapy
**Advanced Imaging Markers**			
**18F-FDG PET/CT uptake intensity**	SUVmax > 3.5: OR 2.8 (95% CI 1.6–4.8)	[84]	Reflects metabolic activity of infected vegetation
**Brain MRI with acute silent infarcts**	OR 2.2 (95% CI 1.3–3.6)	[85]	Indicates ongoing embolization; may warrant early surgery
**Cardiac CT detection of vegetation instability**	89% sensitivity	[67]	Complementary to echocardiography
**Risk Scoring Systems**			
**Italian SEU score ≥ 7 points**	65% risk vs. 5% if <7 points	[16]	Integrates vegetation size, etiology, and underlying conditions
**ENVELOPE score ≥3**	OR 3.5 (95% CI 2.3–5.4)	[84]	Combines echocardiographic and microbiological parameters
**Embolic Risk French Calculator**	86% accuracy	[42]	Web-based tool for clinical use
**Monaldi diagnostic score model**	Low (0–2 points): 22% EE incidence Intermediate (3–5 points): 53% EE incidence High (6–8 points): 78% EE incidence	[69]	Score incorporates the following: 1. *S. aureus* infection (2 points) 2. CRP > 6.7 mg/dL (2 points) 3. Splenomegaly (2 points) 4. Vegetation size ≥ 14 mm (1 point) 5. D-dimer > 747 ng/mL (1 point) Model’s modest discriminative power (LR+ 1.69, LR− 0.33) limits standalone use, necessitating integration with imaging

**Table 4 jcm-14-03965-t004:** Inherited and acquired thrombophilias associated with IE.

Type	Specific Condition	Prevalence in IE	Mechanism in IE Pathogenesis	Clinical Impact	References
**Inherited**	FVL	6.4% (vs. 3.25% in controls)	Enhanced thrombin generation, fibrin deposition	Increased risk in device-related IE	[81]
	FII G20210A	8.3% in prosthetic valve IE	Elevated prothrombin levels	Higher thrombotic complications	
	Protein C/S deficiency	Limited data	Impaired anticoagulant function	Unknown	[141]
**Acquired**	APS	Variable	β2-glycoprotein-I mediated platelet activation	Increased embolic risk	[142]
	Malignancy-associated	Common in NBTE	Predisposition to sterile vegetations	Secondary infection risk	[143,144]
	COVID-19-related	Emerging data	Endothelial damage, NETosis	Increased IE risk in critically ill	[145,146]

**Table 5 jcm-14-03965-t005:** Effect of therapeutic interventions on embolic risk in IE.

Intervention	Timing	Effect on Embolic Risk	Supporting Evidence	Clinical Recommendations
**Antibiotic Therapy**	Early (first 2 weeks)	10–20× higher risk during initiation	[75]	Intensive monitoring during initial therapy
	Later phases	Reduced risk with >40% vegetation size reduction	[149]	Consider early surgery if no size reduction
**Antiplatelet Therapy**	Prior chronic use (≥6 months)	64% reduction (aOR 0.36, 95% CI 0.19–0.68)	[83]	Continue pre-existing therapy
	De novo initiation	No benefit (OR 1.62, 95% CI 0.68–3.86)	[150]	Not recommended (Class III)
**Anticoagulant Therapy**	Pre-admission	Reduced vegetation size (>10 mm)	[86]	Continue if indicated for other reasons
	De novo	No significant embolic reduction, 71% higher hemorrhagic risk	[86]	Avoid unless specifically indicated
**Early Surgery**	Within 48 h	78% mortality reduction (HR 0.22)	[151]	Consider for mobile vegetation >10 mm

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
