# Peer review of "Deciphering the Complex Relationships Between the Hemostasis System and Infective Endocarditis"

_jcm, 2025, doi:10.3390/jcm14113965_

Round 1

Reviewer 1 Report

Comments and Suggestions for Authors

The authors present a comprehensive review of the role hemostasis plays in both the pathogenesis and clinical features of infective endocarditis. Below are some feedback points.  

  1. There seems to be a major issue with the citations, which all need careful review. A number of citations do not match or support the statements in the paper. Additionally, some citations will need reformatting maintain consistency throughout the paper - e.g., paragraphs 3.2, 5.1.
  2. The term "non-bacterial thrombotic endocarditis (NBTE)" is a clinical diagnosis, which encompasses marantic endocarditis and Libman Sacks endocarditis as seen in the setting of malignancy or rheumatological disease. I would avoid using this term to describe the pathogenesis of infective endocarditis as in lines 134 and 142.
  3. line 148. Should "bacterial surface" be "...vegetation surface"? Double check the references - I could not find supporting evidence in the references provided [26,27].
  4. line 175. "annual" should be "one-year" - embolic risk is upfront in the first year and decrements after time. 
  5. Lines 194-196. I am also confused by what is meant by "89% sensitivity for identifying unstable lesions." Again, the references provided [52,57] do not support these statements; please verify if these references are indeed correct. 
  6. In table 2, clarify the following:
    1. S. aureus etiology. What is meant by 35-61% risk? The reference cited indicates an adjusted odds ratio of 1.76 (95% CI 1.09–2.86) using multivariable logistic regression. Evidence suggests that S. aureus is only associated with embolic events before treatment initiation, and does not influence embolic risk after initiation.
    2. Carotid intima-media thickness >1.0 mm. What is meant by "marker of pre-existing atherosclerosis interacting with IE"?
  7. Lines 355-357. "Furthermore, the device-related IE demonstrates stronger associations with inherited hypercoagulable states, as FVL’s interaction with intravascular hardware amplifies fibrin deposition and biofilm formation." needs a reference.
  8. Within the discussion about early surgery to reduce embolic risk (lines 422-431), the authors should strongly consider including discussion of the EASE trial, which is a landmark RCT in endocarditis.
  9. Line 427-428. What is meant by " enhancing vegetation friability?"
  10. The abstract should be re-written for clarity and to better summarize the content matter.

Author Response

Please look at attached file

Reviewer 2 Report

Comments and Suggestions for Authors

The manuscript is comprehensive, well-structured, and addresses an important and complex aspect of Infective Endocarditis pathophysiology.

Here are some suggestions.

  1. While the manuscript extensively discusses molecular mechanisms, the authors should mention how this affects clinical decision-making. Please add a short clinical workflow figure or table summarizing how hemostasis parameters (e.g., D-dimer, PT-INR) should guide therapy decisions.
  2. The immunothrombosis section is well-detailed. However, highlighting therapeutic strategies aimed at modulating immunothrombosis (e.g., NETs inhibitors, FXI inhibitors) earlier in the review could strengthen its clinical impact.
  3. The discussion on antiplatelet and anticoagulant therapy timing is good. Please summarize these findings in a concise figure that depicts treatment decision points over the clinical timeline of IE.
  4. The manuscript mentions differences between S. aureus and other pathogens. A dedicated subsection or a comparative table summarizing vegetation characteristics, embolic risk, and coagulopathy profiles across key pathogens (e.g., S. aureus vs Streptococcus viridans vs fungal IE) would make it better understanding.
  5. Given the manuscript’s focus on platelet-driven mechanisms, I recommend adding additional studies that specifically address platelet heterogeneity and platelet activation dynamics in sepsis and endocarditis.

In particular, please consider citing the following recent paper, which may complement and strengthen your discussion:

Qiu, X., Nair, M. G., Jaroszewski, L., & Godzik, A. (2024). Deciphering Abnormal Platelet Subpopulations in COVID-19, Sepsis and Systemic Lupus Erythematosus through Machine Learning and Single-Cell Transcriptomics. International Journal of Molecular Sciences, 25(11), 5941.

Author Response

Please look at attached file

Round 2

Reviewer 1 Report

Comments and Suggestions for Authors

Thank you for your revisions and for the comprehensive review on this interesting topic. 

Reviewer 2 Report

Comments and Suggestions for Authors

The authors revised accordingly.